# Increased Adherence to the Mediterranean Diet and Higher Efficacy Beliefs Are Associated with Better Academic Achievement: A Longitudinal Study of High School Adolescents in Lebanon

**DOI:** 10.3390/ijerph18136928

**Published:** 2021-06-28

**Authors:** Joyce Hayek, Hein de Vries, Maya Tueni, Nathalie Lahoud, Bjorn Winkens, Francine Schneider

**Affiliations:** 1Department of Health Promotion, School for Public Health and Primary Care (CAPHRI), Faculty of Health Medicine and Life Sciences, Maastricht University, P.O. Box 616, 6200 Maastricht, The Netherlands; hein.devries@maastrichtuniversity.nl (H.d.V.); francine.schneider@maastrichtuniversity.nl (F.S.); 2Department of Biology, Nutrition and Dietetics, Faculty of Sciences II, Lebanese University, Fanar P.O. Box 90656, Lebanon; mayatueni@hotmail.com; 3Pharmacoepidemiology Surveillance Unit, Center for Research in Public Health (CERIPH), Faculty of Public Health, Lebanese University, Fanar P.O. Box 90656, Lebanon; nathalie.lahoud@hotmail.com; 4Department of Methodology and Statistics, School for Public Health and Primary Care (CAPHRI), Faculty of Health Medicine and Life Sciences, Maastricht University, P.O. Box 616, 6200 Maastricht, The Netherlands; bjorn.winkens@maastrichtuniversity.nl

**Keywords:** health behaviors, socio-cognitive factors, academic achievement, adolescents, Lebanon

## Abstract

This longitudinal study aims to examine how changes in health behaviors and socio-cognitive factors influence the academic achievement of Lebanese adolescents over a period of 12 months. Adolescents (*n* = 563) from private and public schools in Mount Lebanon and the Beirut area, aged between 15 and 18, participated in a three-wave longitudinal study and completed a self-administered questionnaire assessing socio-demographics, health behaviors, socio-cognitive factors, parenting styles, and academic achievement. A linear mixed model was carried out to examine if changes in health behaviors and cognitive factors affect changes in academic achievement after 6 and 12 months from the baseline, adjusting for demographic variables and parenting style. Results show that improved adherence to the Mediterranean diet and an increase in self-efficacy were associated with an increase in academic achievement. An increase in adherence to the Mediterranean diet had the same effect on academic achievement 6 and 12 months from the baseline, whereas an increase in efficacy beliefs was only significantly associated with achievement at 12 months from the baseline. This study supports the longitudinal link between diet quality and efficacy beliefs with the academic achievement of adolescents. This relationship is independent of sex, age, religion, parents’ education, and raising styles.

## 1. Introduction

Youth has been recognized as important advocates and agents of change for more sustainable communities and prosperous future societies [1,2]. One way of empowering adolescents is through education [3]. Good education and academic achievement of adolescents have been a growing area of research, as it brings reward to the individual as well as to society as a whole [4]. Academic achievement represents performance outcomes on academic subjects and reflects the acquisition of knowledge and skills, and is usually measured in terms of grades attained [5].

Good academic achievement has been linked to several positive outcomes for the adolescent in both the short and long term. Academically successful adolescents have more chances to get into good universities, which opens up doors to better career opportunities, financial security, and overall better quality of life [6,7,8]. In addition, academic achievement has positive impacts on different social outcomes; academically successful adolescents are more confident, have higher self-esteem [9,10,11], are more engaged citizens, and are less dependent on social assistance [12]. On the other hand, low achievers are more likely to fail and repeat their grades, leave school early, and are less likely to pursue higher education [13]. Consequently, individuals with lower educational attainment have fewer employment opportunities, earn lower wages, pay fewer taxes [14,15], and are more likely to suffer from negative attitudes, lower self-esteem [16], and delinquent involvement [17].

Additionally, educational failure can have negative effects on the country’s economy and growth by limiting productivity and innovation [18]. Despite an improvement in overall literacy rates, school failure and dropouts are still problems faced by low- and high-income countries. The latest data shows that on average, 15% of young adults aged 25 to 34 have not attained upper secondary education [19]. It has also been found that for those individuals, unemployment rates are twice as high compared to those with a university degree [19]. Investigating and understanding determinants of academic achievement is thus imperative to improve academic outcomes and prevent school failure and dropouts, and their adverse consequences for both individuals and society.

Academic performance is associated with several different factors, including non-modifiable factors such as genetic predisposition [20], and partially modifiable factors such as lifestyle and motivational factors [21]. In the present study, the focus is on changeable determinants of academic achievement as they can be targeted and improved in tailored interventions to foster future academic achievement. Among the most important studied changeable determinants of achievement are health behaviors [22]. Healthy behaviors such as being physically active, following a healthy dietary pattern, fruit and vegetable consumption have been linked to good cognitive functioning and better achievements [23,24,25]. In addition, health-risk behaviors such as unhealthy eating, smoking, and alcohol consumption have been found to adversely affect academic achievement [26,27]. In turn, good academic achievement has been found to predict better future health through greater exposure to resources and information and greater health awareness, all of which leads to a healthier lifestyle [28].

Other factors known to influence the academic achievement of adolescents are socio-demographics [29], social-cognitive [30], and environmental factors [31]. Social cognitions such as self-efficacy and intentional behavior are known to influence academic achievement [32]. Adolescents with high efficacy beliefs would put more effort into their academic work and consequently perform better [33,34]. Moreover, a nurturing home environment has also been found to be vital to a child’s education [35,36]. Parents can influence their adolescent offspring through parenting styles [37]. Parenting style has been described as a constellation of two underlying dimensions: responsiveness and demandingness [38]. Based on these two dimensions, a four-typology classification has been identified: authoritative parents are responsive and demanding, authoritarian is demanding but not responsive, permissive are responsive but not demanding, and neglectful are neither responsive nor demanding [39,40]. Findings from previous studies have majorly shown that having an authoritative parent fosters a better outcome for the child which includes academic outcomes [41,42].

Previous research examining predictors of academic performance has focused on one factor or explored the influence of limited subsets of behaviors, instead of including multiple factors comprehensively. Additionally, most of the existing research used a cross-sectional design highlighting the need to conduct more longitudinal research. Academic achievement of adolescents, particularly in secondary school, is crucial as it will set the stage for university entry and better future prospects. In Lebanon, very few studies have investigated the health behaviors of adolescents and only one recent study explored their relation to academic achievement [43]. Moreover, according to the results of international tests, Lebanese adolescents are lagging behind peers from other countries, pointing toward growing disparities in academic performance [44]. Investigating determinants of academic achievement and how they affect it will pave the way for evidence-based interventions to enhance the achievement of adolescents in Lebanon. Existing literature provides preliminary evidence that it is possible to improve school outcomes through improving health behaviors and motivational factors [45,46]. Hence, gaining further insight into those factors will help develop targeted multicomponent intervention programs that are culture-specific to promote academic achievement.

To our knowledge, no prospective study has examined the association between changes in health behaviors and motivational factors and changes in the academic achievement of adolescents while controlling for demographic and environmental factors. Therefore, this study aimed to investigate how changes in health behaviors and motivational factors affect changes in the academic achievement of Lebanese adolescents at a follow-up after 6- and 12-months while controlling for socio-demographics and parental styles. Secondly, this study examined if associations were the same at 6 months from baseline compared to 12 months from baseline.

## 2. Materials and Methods

### 2.1. Design

The study had a longitudinal design: the baseline survey (t1) was administered in Spring 2017, and follow-up surveys were distributed again six months (t2) and 12 months after the baseline (t3). This study was conducted in accordance with the Declaration of Helsinki [47], and the protocol was approved by the Lebanese Ministry of Education and Higher Education (10/684; date: 1 March 2017), and the Al Hayat Hospital ethical committee (ETC112018). Written consent forms were obtained from all students and their parents before participation.

### 2.2. Participants and Procedure

The study was conducted in private and public schools in Beirut and the Mount Lebanon area. The Lebanese Educational system is divided into two sectors: public and private. Public schools are non-profitable, free of charge, and under government authority. Private schools are operated by individuals or organizations with the government having a weaker control and have usually higher tuition fees, making them only accessible to well-off individuals. The educational system is divided into three cycles, elementary intermediate, and secondary. The secondary level is particularly important as it is concluded with official exams, “Lebanese Baccalaureate”, qualifying students for tertiary education. A total of ten schools (five private and five public) were randomly selected from the Ministry of Education’s list of schools based on the stratified sampling design, the strata being public and private schools. From the initial 10 schools approached, seven (four private and three public) agreed to participate in the study. G*Power version 3.1.9.7 was used to compute the sample size. We fixed the study power and the confidence level at 95%. We assumed that a linear multiple regression (fixed model; R2 deviation from zero) with a medium effect size of Cohen of 0.15 would be an appropriate strategy to answer our primary objective (determinants of academic performance taken as a continuous variable). Thus, to study around 20 variables (age, gender, religion, parents’ education, school type, diet, physical activity, smoking, cognitive variables, etc.), a minimum sample size of 222 was required. In order to take failure to follow-up and incomplete questionnaires into account, we recruited over 500 participants to, at a minimum, double the sample size. All students in Grade 10 and 11 (aged 15 to 18) were invited to participate in the survey. The baseline sample was a total of 600 adolescents, out of which 563 (94%) provided valid data. Participants with complete measurements at the six-month follow-up and 12-month follow-up totaled 362 (64.3%) and 345 adolescents (61.3%), respectively. Apart from adolescence being a period of significant development where lasting behaviors are adopted, this particular group was studied because it represents the last years of school (secondary level) prior to entering university, and is considered important and decisive of student’s future academic endeavors. Paper and pencil self-administered questionnaires were filled out by the students in the classroom. The questionnaire collected socio-demographic, lifestyle, and motivational data. Trained dieticians were present for any clarification and to measure students’ height and weight using calibrated equipment [48]. Students were weighed to the nearest 0.1 kg, using a Seca-calibrated electronic weighing scale (Hamburg, Germany) and height was measured to the nearest 0.5 cm by using a portable stadiometer (ADE stadiometer, Germany). All measurements were carried out in light indoor clothes and without shoes. BMI was calculated as weight in kilograms divided by the square of height in meters (kg/m^2^).

### 2.3. Questionnaire

#### 2.3.1. Socio-Demographics

Socio-demographic data included information on students’ sex (1 = male; 2 = female), age (1 = 15; 2 = 16; 3 = 17; 4 = 18), type of school (1 = public; 2 = private), educational level of parents (low = never went to school & primary school; medium = complementary & secondary school; high = technical school & university), family structure (1 = living with both parents; 2 = other arrangements) and religion (0 = Not Christian; 1 = Christian).

#### 2.3.2. Parenting Styles

Parenting styles were assessed using the Authoritative Parenting Index (API) [49]. The API measures the two dimensions of parenting behavior, responsiveness, and demandingness, as perceived by the adolescents. Nine items measured responsiveness (e.g., “She/he listens to what I have to say”) and seven items measured demandingness (e.g., “She/he has rules that I must follow”). In this study, the items were worded in reference to both parents (e.g., “They make sure I go to bed on time”). Students used a four-point response scale (1 = Not like them, 2 = Sort of like them, 3 = A lot like them, and 4 = Just like them) to indicate how closely the statement matches their parents. The final scale scores could range from 9–36 for responsiveness (Cronbach’s α = 0.80) and 7–28 for demandingness (Cronbach’s α = 0.70).

Parenting styles were created using median splits on demandingness and responsiveness. Based on the combination of the levels of responsiveness and demandingness the four parenting styles were categorized as authoritative (high on both responsiveness and demandingness), authoritarian (high demandingness and low responsiveness), permissive (low demandingness and high responsiveness), and neglectful (low on both responsiveness and demandingness). 

#### 2.3.3. Lifestyle Factors

##### KIDMED

Students completed a semi-quantitative Food Frequency Questionnaire (FFQ) that included 64 food and beverage items commonly consumed in Lebanon [50] and food habits questions (breakfast consumption, snacking, and frequency of eating fast food).

The collected data were used to calculate the KIDMED index (Mediterranean Quality Index for children and adolescents) [51]. The KIDMED evaluates adherence to the Mediterranean diet in children and adolescents based on the consumption of 16 items, of which 12 are positively scored and four negatively scored [52]. Items denoting a negative connotation to the Mediterranean diet are assigned a value of −1 (consumption of fast food, baked goods, and pastries for breakfast, consumption of sweets and candies several times a day, and skipping breakfast). Items denoting a concordance to the Mediterranean diet are scored +1 (consumption of fruits, vegetables, fish, pulses, nuts, cereals and grains, dairy products, and olive oil). The total score ranges from 0 (very poor) to 12 (high adherence) [52].

##### Physical Activity

Physical activity (PA) was measured using the short version of the International Physical Activity Questionnaire (IPAQ) [53]. Adolescents provided information on the time spent walking, in moderate and vigorous activity over the past seven days. For each activity intensity, time spent performing the activity is multiplied by the metabolic equivalent of task (MET) estimated at 3.3 for walking, 4.0 for moderate-intensity activity, and 8.0 for vigorous-intensity activity. The total activity score is calculated by summing the MET-minutes/week for each activity. The total score was reported as a continuous measure and converted into MET-hr/day [54,55].

##### Smoking and Alcohol

Prevalence of smoking and drinking in the past 30 days was assessed with the following questions: “During the past month, on how many days did you smoke?” and “During the past month, on how many days did you drink alcohol?” the responses were “0 days; 1 or 2 days; 3 to 5 days; 6 to 9 days; 10 to 19 days; 20 to 29 days; All 30 days”. In line with the reported prevalence used in the Global School-based Student Health Survey, the responses were then dichotomized into (1) no = 0 days and (2) yes = 1–30 days [56,57].

#### 2.3.4. Socio-Cognitive Factors

Four questions assessed adolescents’ favorable and unfavorable attitudes towards getting good academic grades. Two items measured favorable attitudes “Getting good academic grades is a good help for getting a good job/will get me compliments from my parents”. The responses were coded from −2 (strongly disagree) to +2 (strongly agree). Two items assessed unfavorable attitudes “Getting good academic grades means that I have to work too hard/will cause disapproval among my friends” using the same scale reverse coded from +2 (strongly agree) to −2 (strongly disagree) so that higher scores reflect a more favorable attitude towards getting good grade (α = 0.15).

Three questions assessed the subjective norms of mother, father, and teacher on a five-point scale from +2 (strongly agree) to −2 (strongly disagree): ‘My father/my mother/my teacher expects me to get good academic grades’ (α = 0.56).

Five questions assessed the extent to which adolescents thought they were able to get good grades: “I find it easy to get good academic grades/to concentrate at school for getting good academic grades/to master the skills that are taught in class this year/to concentrate on schoolwork when I am at home/to finish all my school work”. Responses ranged from strongly agree (+2) to strongly disagree (−2) (α = 0.76).

One statement assessed the intention to get good academic grades: ‘I intend to get good academic grades’ on a five-point Likert scale from +2 (strongly agree) to −2 (strongly disagree) [58].

#### 2.3.5. Outcome Measure: Academic Performance

Academic achievement was measured by asking adolescents to report their general average which is the result of the performance of the student in all school subjects during a specific semester. The general average is the standard instrument for the assessment of the academic achievement of students in Lebanese schools [43]. All participating schools use a 0–20 scale where the passing grade is 10 out of 20.

### 2.4. Statistical Analysis

Data were analyzed using IBM SPSS Statistics for Windows, version 23 (Armonk, NY, USA: IBM Corp.). Mean ± standard deviation (SD) was used to present numerical variables, while the number of participants (%) was used for categorical variables. Changes from baseline were assessed using the McNemar–Bowker test for categorical variables and paired-samples t-test for numerical variables. Normality was checked using a histogram.

The first research question was to examine how the change in our independent variables (health behaviors and socio-cognitive variables) affect the change in our outcome variable (academic performance) between baseline and 6 months (∆t2) and baseline and 12 months (∆t3). To account for repeated measures two-level linear mixed model analyses were performed with repeated measurements as the first level and adolescents as the second level, where an unstructured covariance structure was considered for the repeated measures. No missing data were imputed as the likelihood-based approach was used to deal with missing outcome data. This approach assumes the missingness to be at random (MAR).

An attrition analysis was performed to check which variables were related to missingness using logistic regression analysis; variables related to missingness were included in the linear mixed model analyses. The fixed part of the model consisted of the socio-demographics, parenting style, change scores of health behaviors and socio-cognitive variables, and time (∆t2, ∆t3), where interactions between these change scores and time were included to examine the second research question, i.e., whether the effect of these change scores is different after 6 months (∆t2) than after 12 months (∆t3). A top-down procedure was performed to assess the significance of the interaction terms. In case an interaction term was statistically significant, the effect of the corresponding variable was reported for both time points separately. Otherwise, the interaction term was removed from the model, where the effect at both time points was combined into one effect estimate. The effect estimates were reported with 95% confidence intervals and *p*-values, where a two-sided *p*-value ≤ 0.05 was deemed as statistically significant.

## 3. Results

### 3.1. Demographics

The study sample consisted of 49.7% male and 50.3% female participants, with a mean age of 16.8 years. Out of the study participants, 62.5% of subjects attended private school. The proportion of parents with a high educational level was 52.5% for fathers and 55.1% for mothers. Adolescents had low adherence to the Mediterranean diet (mean ± SD: 3.76 ± 2.49). In all, 9.2% of adolescents reported smoking in the previous month and 54.7% reported drinking (Table 1).

### 3.2. Change in Health Behaviors, Socio-Cognitive Factors, and Academic Achievement at ∆t2 and ∆t3

At ∆t2 an increase in the mean score of the following variables was reported for the study population: Academic achievement (0.51 ± 1.19), KIDMED (1.30 ± 2.13), BMI (0.25 ± 1.49), and intention (0.06 ± 0.95). A decrease was reported for physical activity (−1.02 ± 5.28), attitude (−0.02 ± 0.50), social norms (−0.007 ± 0.70), and self-efficacy (−0.18 ± 0.67). At ∆t3 the following variables showed an increase in the mean score: academic achievement (0.35 ± 1.49), KIDMED (1.58 ± 2.25), BMI (0.31 ± 1.72), whereas a decrease was reported for physical activity (−0.47 ±6.18), attitude (−0.06 ± 0.52), social norms (−0.012 ± 0.70), self-efficacy (−0.13 ± 0.73) and intention (−0.01 ± 0.93). At both ∆t2 and ∆t3, the majority of adolescents reported to be still not smoking (89.5% and 87.5% respectively) whereas most adolescents were reported to still be drinking (59.6 at ∆t2 and 59.9% at ∆t3) (Table 2).

### 3.3. Effect of Change in Socio-Cognitive Variables and Health Behaviors on Academic Achievement

Our results show that the interaction between change scores of our independent variables (health behaviors and socio-cognitive variables) and time was only statistically significant for self-efficacy (*p* < 0.001), indicating different effects of self-efficacy to achieve good grades on the outcome at 6 months and 12 months after baseline. At 12-month follow-up we found that higher levels of self-efficacy to achieve good grades were related to higher academic achievement after correction for baseline; for every additional unit increase in self-efficacy, academic achievement additionally increased by 0.48 (effect estimate (B) 0.48; 95% CI 0.32, 0.65). Yet, at 6 months we did not find this effect (B 0.13, 95% CI −0.04, 0.30) (Table 3). Attitude, social norms, and intention change were not significantly related to change in academic achievement (*p* ≥ 0.364).

A significant effect of adhering to the Mediterranean diet on academic achievement was found. As no time interaction was found this implies that this pattern was similar at 6 and 12 months follow-up and therefore combined into one effect estimate. An increase in the adherence to the Mediterranean diet was significantly associated with an increase in academic achievement (B 0.25; 95% CI 0.19, 0.30). Changes in BMI, PA, alcohol, and smoking status did not significantly influence change in academic achievement (*p* ≥ 0.175) (Table 3).

## 4. Discussion

The purpose of this study was to examine how changes in health behaviors and motivational factors affect academic achievement in a sample of adolescents from Lebanon using a longitudinal approach. After controlling for demographic factors (age, gender, school type, family structure, religion, and parents’ education) and parenting style, adherence to the Mediterranean diet and efficacy beliefs were significantly associated with academic performance of Lebanese adolescents after 6- and 12-months follow-up. More specifically, an increase in diet quality and efficacy beliefs was associated with an increase in adolescents’ academic achievement. Changes in PA, BMI, smoking, alcohol status, attitude, social norm, and intention were not statistically significantly associated with a change in academic achievement.

With regards to health behaviors, our findings show that adolescents with improved adherence to the Mediterranean diet had a better academic achievement 6- and 12-months later. Similar findings were reported in previous studies of the longitudinal association of diet with academic performance [59]. In past research, favorable dietary intake characterized by consumption of fruits and vegetables, home-made meals, nuts, and adequate vitamin intake was found to predict greater academic performance [59,60,61], while a Western dietary pattern and consumption of nutrient-poor and refined food were predictive of poorer performance [62,63,64,65]. Our results are also in accordance with findings of previous studies that have explored adherence to the Mediterranean diet and academic achievement among Lebanese high school adolescents [43,66] and university students [67] showing that higher adherence was related to higher academic scores. The Mediterranean diet is considered one of the healthiest eating patterns and higher adherence has consistently been linked with better overall health and cognitive functioning [68,69,70,71]. Adolescents with higher adherence and healthier eating habits are assumed to have parents with higher dietary knowledge [72]. Parents can influence their children’s intake as they are considered the main meal providers and can control access to healthy food, they are also considered role models and can also influence their children’s food attitudes and preferences [73,74]. Adolescents whose parents are involved in their diet are also more likely to be involved in their academic work, which is another factor affecting performance [75]. Moreover, healthy eating behaviors were found to be associated with conscientiousness, which can also lead to better achievement [76]. Conscientious individuals are more organized, efficient, dutiful, and show self-discipline, which positively influences academic performance [77]. Consequently, adolescents with higher adherence to the Mediterranean diet might be applying themselves better to their studies which consequently results in greater achievement. Our findings add to the evidence on the importance of adhering to the Mediterranean diet for achieving good academic performance. Future intervention studies are needed to test the causal effect of associations between the Mediterranean diet and academic achievement.

Even though in prior research, significant associations of smoking and drinking with academic achievement have been reported in cross-sectional and longitudinal studies [27,78,79], in this current study, change in smoking and alcohol status were not significantly related to change in academic achievement. The latter could be due to the time lapse between follow-ups; most longitudinal research reporting a significant influence of substance use on academic performance was conducted over three or more years [79,80] and thus, significant results might be observed over longer follow-up periods. An additional point to consider is that social desirability bias is common among adolescents when reporting on substance use [81], which might explain the null finding. Similarly, our study did not find a significant association between changes in BMI and physical activity and changes in academic achievement, which is in agreement with previous research [82,83,84]. Our results show that an increase in BMI was associated with a decrease in academic achievement, whereas an increase in physical activity was associated with an increase in achievement. However, those associations were not significant. Evidence regarding physical activity, BMI, and academic achievement remains inconsistent with some studies finding a positive association [85,86] and others null associations [82,83,84]. The non-significance could be explained by the insufficient length of the follow-up period to yield significant results and the level of adjustment for confounders. Previous studies that found a positive association between changes in weight status and academic performance were over a two-year period [86].

With respect to cognitive factors, only an increase in self-efficacy was found to be significant in its effect on academic achievement. A possible explanation for this finding is that the change in attitude and social norms was smaller compared to self-efficacy (Table 2), consequently, the amount of change might not have been sufficiently large to elicit a change in academic achievement. Attitude, social norms, and self-efficacy are constructs derived from social cognitive theories such as the I-Change Model and The Theory of Planned Behavior [58,87] that are used to explain why people engage in certain behaviors. According to those theories, the relative importance of each of those constructs may vary across behaviors, populations, and situations [87]. Hence, in our sample self-efficacy could be a stronger predictor of academic achievement, attenuating the influence of attitude and social norms and might be more effective in causing change. Our findings show that a positive change in academic self-efficacy was positively related to higher academic achievement at both 6- and 12-months follow-up, but the effect was significant at 12 months only. Our results are in accordance with prior longitudinal studies that showed that higher efficacy beliefs positively predict academic achievement [88,89,90]. Efficacious adolescents are more likely to perform higher because their strong beliefs about their ability to perform push them to put more effort into their academic work, persevere longer in the face of failure, and ultimately achieve higher, whereas low beliefs lead to less engagement, less effort and perseverance and consequently lower success [34,91,92]. Moreover, our finding of a significant effect of self-efficacy change at only one-year follow-up suggests that the time-point of measurement influences the strength of the relationship between self-efficacy and performance. Self-efficacy change better correlates with achievement after a period of time, in our case one year, and may not strongly relate with achievement when measured at short time-points apart. A possible explanation to why increased self-efficacy has a stronger positive effect in the longer term may result from the potential bi-directional relation between self-efficacy and performance [93]. When adolescents see that hard work and persistence are paying off by them achieving progress in academic courses, this successful academic experience will, in turn, further foster their efficacy beliefs which will consecutively contribute to them achieving more. Thus, self-efficacy beliefs become stronger predictors of academic achievement as adolescents progress through the academic year.

Finally, change in intention was not significantly associated with a change in academic achievement. The latter could be due to habit moderating the intention-behavior relation [94]. The impact of intention on behavior might differ based on the extent to which the behavior is habitual [94]. In other words, adolescents might have certain study habits or patterns of studying that occur without conscious effort and which could also be affecting performance [95,96], and thus the impact of intention on academic achievement change is reduced. Research suggests that the intention-behavior relation is dependent on implementation intention and goal setting [97], forming if-then plans or implementation intentions could help overcome the control of habit [98].

## 5. Limitations

A few limitations should be considered in the interpretation of the findings. First, grades were self-reported and the possibility of respondent bias cannot be ruled out. However, previous research indicates that self-reported grades can be reliable and correlate strongly with accurate grades [99,100]. In addition, our sample was selected from only two regions in Lebanon, consequently, generalization of the results should be carried out with caution. Furthermore, to increase the accuracy of predictions, self-efficacy at task-specific and context-specific levels should be considered in future research. Moreover, while standard measures to assess socio-economic status are available [101], we opted to select those that were the easiest to answer. Yet, future studies may be needed to identify the most optimal way for assessing socio-economic status in study populations such as ours. Another limitation was the low Cronbach α for attitude, however, sensitivity analysis (excluding change in attitude from the model) gave similar results (data not shown). Finally, even though the study controlled for the influence of a variety of factors such as age, gender, school type, religion, parents’ education, and parenting style, the unique role of other important variables related to cognitive ability and specific learning disorders may still need to be considered in future research. Despite the above caveats, this is the first study that we are aware of to analyze prospective associations between adolescents’ dietary patterns and efficacy beliefs with academic achievement in Lebanon and the MENA region. This study expanded our understanding of how a change in lifestyle and motivational factors could affect academic achievement among high school adolescents. An additional strength of this study is the methodological approach used. Major advantages of using linear mixed model analyses are that it uses all available data (no list-wise deletion), accounts for the correlation between repeated measures, and assumes missing outcome data to be missing at random (missingness might depend on observed variables, which were therefore included in the model).

## 6. Conclusions

In conclusion, the findings of this study suggest that an improvement in adherence to the Mediterranean diet and an increase in self-efficacy beliefs were associated with an increase in academic achievement during a one-year period. This implies that education intervention programs should promote the adoption of a healthy dietary pattern and increase perceptions of self-efficacy in order to enhance adolescents’ chances of achieving higher. Future research should be directed towards interventions that examine whether experimental manipulation of health behaviors and socio-cognitive factors results in a corresponding change in academic achievement.

## Figures and Tables

**Table 1 ijerph-18-06928-t001:** Descriptive characteristics of the baseline sample (N = 563).

Characteristics	Frequency (%)
Type of school	
Public	211 (37.5)
Private	352 (62.5)
Gender	
Male	283 (50.3)
Female	280 (49.7)
Religion	
Christian	435 (77.3)
Non-Christian	128 (22.7)
Family Structure	
Living with both parents	507 (90.1)
Other arrangements	56 (9.9)
Father education	
Low	31 (6.5)
Medium	195 (41.0)
High	250 (52.5)
Mother education	
Low	22 (4.4)
Medium	204 (40.6)
High	277 (55.1)
Age	
15	232 (41.2)
16	227 (40.3)
17	85 (15.1)
18	19 (3.4)
Mean (±SD)	16.76 ± 0.73
Parenting Style	
Authoritative	107 (31.4)
Authoritarian	100 (29.3)
Permissive	68 (19.9)
Neglectful	66 (19.4)
Smoking	
No	511 (90.8)
Yes	52 (9.2)
Alcohol	
No	255 (45.3)
Yes	308 (54.7)
PA	
Mean (±SD)	6.20 ± 6.98
BMI	
Mean (±SD)	23.6 ± 4.50
KIDMED	
Mean (±SD)	3.76 ± 2.49
Attitude	
Mean (±SD)	0.59 ± 0.47
Social norm	
Mean (±SD)	0.95 ± 0.67
Self-efficacy Total	
Mean (±SD)	0.25 ± 0.73
Intention	
Mean (±SD)	1.15 ± 0.90
Academic achievement	
Mean (±SD)	12.05 ± 2.45

**Table 2 ijerph-18-06928-t002:** Mean change scores of health behaviors and socio-cognitive factors of Lebanese adolescents at the 6 and 12 months follow-up.

Characteristics	∆t2	*p*	∆t3	*p*
**Smoking**				
Stopped smoking	6 (1.7)		5 (1.5)	
Still smoking	17 (4.7)	0.005 ^a^	16 (4.7)	<0.001 ^a^
Started smoking	21 (5.8)		27 (7.9)	
Still not smoking	317 (87.8)		295 (86)	
**Alcohol**				
Stopped drinking	29 (8.0)		30 (8.7)	
Still drinking	187 (51.8)	1.000 ^a^	173 (50.3)	0.708 ^a^
Started drinking	28 (7.8)		33 (9.6)	
Still not drinking	117 (32.4)		108 (31.4)	
**PA**				
Mean (±SD)	−1.02 ± 5.28	<0.001 ^b^	−0.47 ± 6.18	0.152 ^b^
**BMI**				
Mean (±SD)	0.25 ± 1.49	0.002 ^b^	0.31 ± 1.72	0.001 ^b^
**KIDMED**				
Mean (±SD)	1.30 ± 2.13	<0.001 ^b^	1.58 ± 2.25	<0.001 ^b^
**Attitude**				
Mean (±SD)	−0.02 ± 0.50	0.450 ^b^	−0.06 ± 0.52	0.018 ^b^
**Social norms**				
Mean (±SD)	−0.007 ± 0.70	0.842 ^b^	−0.012 ± 0.70	0.741 ^b^
**Self-efficacy Total**				
Mean (±SD)	−0.18 ± 0.67	<0.001 ^b^	−0.13 ± 0.73	<0.001 ^b^
**Intention**				
Mean (±SD)	0.06 ± 0.95	0.170 ^b^	−0.01 ± 0.93	0.818 ^b^
**Academic achievement**				
Mean (±SD)	0.51 ± 1.19	<0.001 ^b^	0.35 ± 1.49	<0.001 ^b^

Notes: ^a^
*p*-value for the McNemar-Bowker test, ^b^
*p*-value for the paired-samples *t*-test.

**Table 3 ijerph-18-06928-t003:** Six and 12 months estimated change scores effects on the academic performance of Lebanese adolescents.

Variables	β	95% CI	*p*
**Type of school**			
Public	−0.13	[−0.49, 0.23]	0.473
Private	0		
**Gender**			
Male	−0.06	[−0.36, 0.22]	0.650
Female	0		
**Religion**			
Not Christian	0.08	[−0.43, 0.61]	0.735
Christian	0		
**Family Structure**			
Living with both parents	−0.05	[−0.63, 0.53]	0.858
Other arrangements	0		
**Father’s educational level**			0.268
Low	−0.46	[−1.06, 0.12]	0.124
Moderate	0.002	[−0.29, 0.29]	0.989
High	0		
**Mother’s educational level**			0.663
Low	0.27	[−0.45, 1.10]	0.493
Moderate	−0.06	[−0.36, 0.24]	0.699
High	0		
**Age**			0.006
15	1.52	[0.54, 2.51]	0.003
16	1.23	[0.27, 2.19]	0.012
17	1.55	[0.58, 2.52]	0.002
18	0		
**Parenting style**			0.795
Neglectful	−0.03	[−0.43, 0.40]	0.886
Permissive	−0.18	[−0.56, 0.23]	0.351
Authoritarian	−0.10	[−0.44, 0.26]	0.564
Authoritative	0		
**Smoking status change**			0.701
Stopped smoking	−0.37	[−1.19, 0.43]	0.361
Still smoking	−0.27	[−0.92, 0.37]	0.408
Started smoking	0.08	[−0.34, 0.51]	0.708
Still not smoking	0		
**Alcohol status change**			0.444
Stopped drinking	0.13	[−0.28, 0.55]	0.524
Still drinking	−0.05	[−0.38, 0.27]	0.744
Started drinking	0.20	[−0.19, 0.59]	0.322
Still not drinking	0		
**PA Change**	0.01	[−0.01, 0.03]	0.175
**BMI Change**	−0.01	[−0.08, 0.05]	0.651
**KIDMED change**	0.25	[0.19, 0.30]	<0.001
**Attitude change**	0.86	[−0.10, 0.27]	0.364
**Social norms change**	−0.02	[−0.16, 0.12]	0.780
**Self-efficacy change**	0.14	[−0.03, 0.32]	0.113
∆t2∆t3	0.48	[0.31, 0.64]	<0.001 *
**Intention change**	−0.02	[−0.12, 0.08]	0.686

β = Effect size; CI = confidence interval. Dependent variable: Academic Achievement. ∆t2 = t2 − t1, where t1 = baseline and t2 = 6 months after baseline. ∆t3 = t3 − t1, where t1 = baseline and t3 = 12 months after baseline Association was significant: *p* < 0.05 * Effects significantly different (*p* < 0.001).

## Data Availability

The data presented in this study are available upon reasonable request.

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
