# Peer review of "Increased Adherence to the Mediterranean Diet and Higher Efficacy Beliefs Are Associated with Better Academic Achievement: A Longitudinal Study of High School Adolescents in Lebanon"

_ijerph, 2021, doi:10.3390/ijerph18136928_

Round 1

Reviewer 1 Report

This is an interesting paper dealing with the factors potentially connected with academic achievement. The study is well conducted and reported.

I have only a few points to raise.

1) Was distribution of variables formally assessed? Or why not?

2) Was an a-priori sample sized calculated? Or why not?

3) I'm not sure that the questions used are detailed enough to describe such a complex psychological issue as "self-efficacy". I think the authors should add some comment on this.

4) Were other factors related to academic achievement (e.g. cognitive functioning, Specific Learning Dirsorders...) controlled for? How? Why not?

5) Why not using a standardized assessment of SES, rather than an ad hoc rating? E.g. authors might have used the 4-factors-index by Hollingshead or other similar tools.

6) At the end of the manuscript, the "Patent" section is useless as well as that for "Supplementary Materials" or for "Acknowledgements".

Author Response

Point 1: This is an interesting paper dealing with the factors potentially connected with academic achievement. The study is well conducted and reported. I have only a few points to raise.

Response 1: Thank you. We hope we have addressed all points raised below.

Point 2: Was distribution of variables formally assessed? Or why not?

Response 2: Thank you for this comment. All variables were indeed checked for normal distribution by use of histogram and based on the results we chose the correct statistical test. The following was added in the Statistical Analysis section:

“Normality was checked using histogram.”

Point 3: Was an a-priori sample sized calculated? Or why not?

Response 3: Thank you for this comment. This paper is part of a research project and we did calculate a sample size at the start. We have now addressed this point by adding the below under Participants and procedure:

“ G*Power version 3.1.9.7 was used to compute the sample size. We fixed the study power and the confidence level at 95%. We assumed that a linear multiple regression (fixed model; R2 deviation from zero) with a medium effect size of Cohen of 0.15 would be an appropriate strategy to answer our primary objective (determinants of academic performance taken as continuous variable). Thus, to study around 20 variables (age, gender, religion, parents’ education, school type, diet, physical activity, smoking, cognitive variables etc.), a minimum sample size of 222 was required. In order to take loss to follow up and incomplete questionnaires into account, we decided to recruit more than 500 participants to minimum double the sample size.”

Point 4: I'm not sure that the questions used are detailed enough to describe such a complex psychological issue as "self-efficacy". I think the authors should add some comment on this.

Response 4: We agree with the reviewer that self-efficacy is a complex and broad construct to measure. In our study, in an effort to achieve higher specificity, we tried to refine it and measure academic self-efficacy or self-efficacy to get good grades. We also added the following to the limitations:

“ Furthermore, to increase accuracy of predictions, self-efficacy at task-specific and context-specific level should be considered in future research”

Point 5: Were other factors related to academic achievement (e.g. cognitive functioning, Specific Learning Dirsorders...) controlled for? How? Why not?

Response 5: Thank you for this comment. We did aim to have a comprehensive design with as many confounders as possible. We tried to focus on known confounders of the health behaviors-academic achievement and socio-cognitive-academic achievement relationship. We now mention this limitation in our manuscript:

“Finally, even though the study controlled for the influence of a variety of factors such as age, gender, school type, religion parents’ education and parenting style, the unique role of other important variables related to cognitive ability and specific learning disorders may still need to be considered in future research.”

Point 6: Why not using a standardized assessment of SES, rather than an ad hoc rating? E.g. authors might have used the 4-factors-index by Hollingshead or other similar tools.

Response 6: This is a good point to raise. While we do know that there are several ways to measure SES and researchers tend to make different choices on this matter, we tried to include in our study the most commonly asked questions to measure demographic characteristic in the Lebanese adolescent population (Nabhani-Zeidan, M.; Naja, F.; Nasreddine, L. Dietary Intake and Nutrition-Related Knowledge in a Sample of Lebanese Adolescents of Contrasting Socioeconomic Status. Food Nutr Bull 2011, 32, 75–83). Since the survey was directed at adolescents, we tried to include questions that are easy to answer and tried to avoid questions that might pose a problem or lead to missing or invalid data (such as parents’ income). We feel we already have good variables that are linked to socio-demographics, especially school type which is often considered as a great indicator of SES in the Lebanese context since private schools in Lebanon have high tuition fees making them only accessible to families with higher income whereas families from lower SES background income tend to enroll their children in public schools. We now briefly address this in the limitation:

Moreover, while standard measures to assess socio-economic status are available (Hollingshead, 1975), we opted to select those that were the most easy to answer. Yet, future studies may be needed to identify the most optimal way for assessing socio-economic status in study populations such as ours.”

Point 7: At the end of the manuscript, the "Patent" section is useless as well as that for "Supplementary Materials" or for "Acknowledgements".

Response 7: Thank you for this suggestion. Those empty sections were removed.

Reviewer 2 Report

Dear Authors,

you have presented interesting paper about high school adolescents, but I have some minor and major comments and suggestions.

In Abstract, it will be more interesting for Readers and other Researchers if you will present all research tools you used in your study,

Section Materials and Methods

- please add all information about final number of schools, participants, because in section Results should be presented only final information,

- please add some information how the education in Lebanon looks like - what kind of schools and classes, when they finish and can go to university - it will help Readers to better understand your study,

- add information about Cronbach’s α to research tools

Conclusions are very long could make them shorter.

References have to be verified and modified due to Instruction for Authors of the Journal.

Author Response

Point 1: Dear Authors, you have presented interesting paper about high school adolescents, but I have some minor and major comments and suggestions.

Response 1: Thank you. We hope we have addressed all suggestions below.

Point 2: In Abstract, it will be more interesting for Readers and other Researchers if you will present all research tools you used in your study.

Response 2: We appreciate the reviewer’s suggestion and we agree that adding more information on the scales used in the abstract would be interesting, but we are constrained by the word count (max of 200). All research tools are extensively described under Materials & Methods.

Point 3: Section Materials and Methods, please add all information about final number of schools, participants, because in section Results should be presented only final information.

Response 3: Thank you for this suggestion. We have now moved the text below from Results to the Materials and Methods section:

From the initial 10 schools approached, seven (four private and three public) agreed to participate in the study.
The baseline sample was a total of 600 adolescents out of which 563 (94%) with valid data. Participants with complete measurements at six-month follow-up and 12-month follow-up totalled to 362 (64.3%) and 345 adolescents respectively (61.3%).”

Point 4: please add some information how the education in Lebanon looks like - what kind of schools and classes, when they finish and can go to university - it will help Readers to better understand your study.

Response 4: Thank you, we agree with the reviewer’s point. We have now added the following under Participants and procedure :

The Lebanese Educational system is divided into two sectors: public and private. Public schools are non-profitable, free of charge and under government authority. Private schools are operated by individuals or organizations with the government having a weaker control and have usually higher tuition fees making them only accessible to well off individuals. The educational system is divided in three cycles, elementary intermediate and secondary. The secondary level is particularly important as it is concluded with official exams “Lebanese Baccalaureate” qualifying students for tertiary education.”

Point 5: add information about Cronbach’s α to research tools.

Response 5: Thank you for this comment. Cronbach’s α are now added to all scales. Regarding the low Cronbach α for attitude we performed a sensitivity analysis where the variable change in attitude was removed from the model to see whether the same conclusions are obtained. We had the same results which indicates that the conclusions are robust against including or excluding this variable. The following was added to the limitations:

“ Another limitation was the low Cronbach α for attitude, however sensitivity analysis (excluding change in attitude from the model) gave similar results (data not shown)"

Point 6: Conclusions are very long could make them shorter.

Response 6: Thank you for this suggestion. The Conclusions section was shortened to the following:

In conclusion, the findings of this study suggest that an improvement in the adherence to the Mediterranean diet and an increase in self-efficacy beliefs were associated with an increase in academic achievement during a one-year period. This implies that education intervention programs should promote the adoption of a healthy dietary pattern and increase perceptions of self-efficacy in order to enhance adolescents’ chances of achieving higher. Future research should be directed towards interventions that examine whether experimental manipulation of health behaviors and socio-cognitive factors results in a corresponding change in academic achievement.”

Point 7: References have to be verified and modified due to Instruction for Authors of the Journal.

Response 7: Thank you for this comment. We have worked on the references as per the journal’s guidelines.